# Sugar Intake among Preschool-Aged Children in the Guelph Family Health Study: Associations with Sociodemographic Characteristics

**DOI:** 10.3390/children10091459

**Published:** 2023-08-27

**Authors:** Anisha Mahajan, Jess Haines, Jessica Yu, Gerarda Darlington, Andrea C. Buchholz, Alison M. Duncan, David W. L. Ma

**Affiliations:** 1Department of Human Health and Nutritional Sciences, University of Guelph, Guelph, ON N1G 2W1, Canada; anisha@uoguelph.ca (A.M.); jyu10@uoguelph.ca (J.Y.); amduncan@uoguelph.ca (A.M.D.); 2Department of Family Relations and Applied Nutrition, University of Guelph, Guelph, ON N1G 2W1, Canada; jhaines@uoguelph.ca (J.H.); abuchhol@uoguelph.ca (A.C.B.); 3Department of Mathematics and Statistics, University of Guelph, Guelph, ON N1G2 W1, Canada; gdarling@uoguelph.ca

**Keywords:** sugar intake, children, preschooler, toddler, parent, sociodemographic

## Abstract

**Simple Summary:**

The sociodemographic characteristics of a parent and child may predispose young children to excessive dietary sugar intake; however, there is limited research available on this topic. The aim of this cross-sectional analysis was to examine the impact of sociodemographic characteristics such as child age, child sex, child ethnicity, parent number of years living in Canada, annual household income, parent education and parent marital status on total, free and added sugar intake in young children. Dietary and sociodemographic data including 267 children from 210 families participating in the Guelph Family Health Study were examined. The results indicate that child age, child ethnicity and annual household income can play a crucial role in shaping sugar intake in young children. This research may help inform future research and program interventions early in life along with guiding parents to decrease sugar intake in young children.

**Abstract:**

Background: It is crucial to develop strategies targeted to promote healthy eating patterns in vulnerable populations, especially young children from diverse sociodemographic groups. Thus, the study objective was to investigate the associations between child age, child sex, child ethnicity, parent number of years living in Canada, annual household income, parent education and parent marital status with total, free and added sugar intakes in young children. Methods: This cross-sectional study was a secondary analysis of data gathered in the Guelph Family Health Study. The study included 267 children (129M; 138F) from 210 families aged 1.5 to 5 years. Parents completed questionnaires for children on sociodemographic characteristics and an online 24-hour diet recall. The associations between sociodemographic characteristics and sugar intakes were determined using generalized estimating equations applied to linear regression models. Results: The mean age of the children was 3.5 ± 1.2 years (mean ± std dev.). As children’s age increased, there was a greater intake of free and added sugar ($\hat{\beta }$ = 8.6, *p* = 0.01, 95% CI = 2.4 to 14.7 and $\hat{\beta }$ = 6.5, *p* = 0.03, 95% CI = 0.8 to 12.2, respectively). Those children who identified as white had a higher total sugar intake than children of other ethnicities ($\hat{\beta }$ = 31.0, *p* = 0.01, 95% CI = 7.2 to 54.7). Additionally, higher annual household income was associated with lower was free sugar intake in children ($\hat{\beta }$ = −2.4, *p* = 0.02, 95% CI = −4.5 to −0.4). Conclusions: This study underscores the significant influence of multiple sociodemographic characteristics on sugar intake in young children, providing valuable insights for public health policy and nutrition interventions. Moreover, this study highlights the need for early behaviour interventions focusing on reducing sugar intake in young children, while considering sociodemographic factors.

## 1. Introduction

Overconsumption of sugar intake in children can diminish diet quality, contribute to high energy intake, and lead to excess weight gain and dental caries [1,2]. Research has indicated that preschool-aged children exceed World Health Organization (WHO) guidelines for free sugar intake [1] and consume free and added sugars from a variety of sources [3,4]. Excess sugar intake can contribute to an increased risk of chronic conditions such as type 2 diabetes and metabolic syndrome later in life [5].

It is widely recognized that early life intervention strategies are critical for establishing healthy eating habits that can continue throughout life. Investigating the various environmental factors that can contribute to the establishment of dietary patterns is a complex process. Young children’s dietary intake is influenced by numerous factors at the level of the child, parent, household, and place of childcare [5,6]. In high-income countries, sociodemographic characteristics, especially of the parents, significantly impact the diet quality of children [7]. Research has found that energy intake is higher among children from families with low maternal education, parental unemployment and a greater number of siblings [8]. Furthermore, individuals of lower socioeconomic status (SES) may have increased chronic disease burden later in life [9]. A scoping review on socioeconomic determinants of food intake patterns in young children (2 to 5 years) highlighted that the current knowledge base in this area for young children is sparse [7]. Emerging evidence highlights that diet-related disparities attributed to sociodemographic characteristics may appear in children as early as 4 years of age [10]. Investigating sociodemographic characteristics of both children and parents in association with intake of sugar in young children may help inform early life diet interventions, nutrition messaging to parents and the development of nutrition policy focused on dietary sugars [6,11,12].

There is limited research on how dietary intake of sugar among young children may differ by sociodemographic characteristics. Income and parental education are major indicators driving high sugar intake in many families that have been extensively studied [13]. However, there is heterogeneity in the results and a wide range of other sociodemographic characteristics that have been shown to influence sugar intake including child ethnicity, child immigration status, child age and sex. A German study found that the intake of sweets (including gummies, chocolate and cookies) was higher among preschool-aged children (ages 3 to 6 years) whose families had immigrated to Germany and was inversely associated with parent education level [14]. Similarly, a study from the United Kingdom found an association between toddlers (3 years) consuming sweets, biscuits and chocolate with low maternal education levels and higher financial difficulty [15]. Another study conducted in Canada in children aged 4 and 5 years (*n* = 2114), observed that child sex and age were important considerations when investigating sociodemographic differences in sugar intake in young children [8]. Specifically, boys from low SES consumed higher servings of regular soft drinks when compared to girls; also, children aged 5 years were more likely to regularly consume soft drinks when compared with children aged 4 years [8]. Considered together, these studies present a multifaceted view of numerous sociodemographic factors at both the parental and child level that shape sugar intake in young children.

Most studies have focused on sugar-sweetened beverage (SSB) consumption in relation to the sociodemographic characteristics of an individual [7,8]. Given the wide range of sources of sugar in children’s diets beyond just SSBs, exploring overall sugar intake, including free and added sugars, may provide a more complete picture of the associations between sociodemographic characteristics and children’s sugar intake. Moreover, there is high variability in the sociodemographic characteristics examined to date, where some studies have focused solely on parent-level determinants whereas others have explored child-level determinants [7]. Thus, it is crucial to examine multiple sociodemographic characteristics, including both child and parent-level characteristics, to understand their impact on dietary patterns, specifically sugar intake in young children. These results may help to identify which families to target for interventions designed to reduce children’s sugar intake. Therefore, this study investigated the associations between child and parent sociodemographic characteristics and intakes of total, free and added sugar among preschool-aged and young children.

## 2. Materials and Methods

### 2.1. Study Design, Setting, Participants and Enrollment

This cross-sectional study used baseline data from the Guelph Family Health Study (GFHS), an ongoing family-based obesity prevention intervention [16]. This study began in 2017 and is ongoing and includes 246 families at baseline [16]. Participants met the inclusion criteria for the study if they were residing within the Guelph-Wellington area and were not planning to move within one year, had at least one child 1.5 to 5 years old, and were able to complete a survey in English [16]. Families were recruited through Family Health Teams, Community Health Centres and Ontario Early Years Centres using social media posts, posters and in-person recruitment [16]. The families were given grocery gift cards to recognize the time for participating in the study. The study was approved by the University of Guelph Research Ethics Board (REB#17-07-003).

### 2.2. Data Collection

#### 2.2.1. Dietary Assessment

Dietary data were obtained through one 24-hour recall using the Automated Self-Administered 24-hour Dietary Assessment Tool (ASA24)-Canadian version completed by one parent [17]. Parents of the children completed the 24-hour diet recall online and were instructed to complete the recall for a typical day for their child(ren). Children that were breastfed (*n* = 15), had incomplete records (*n* = 28) or errors in their data (*n* = 12) were excluded from the overall analysis. The final analytic sample included 267 preschool-aged children (138 females; 129 males) from 210 families.

#### 2.2.2. Outcome Variables (Total, Free and Added Sugar)

The outcome variables included baseline total, free and added sugar intakes of preschool-aged children.

##### Total, Free and Added Sugar Intake Calculations

This study included a comprehensive evaluation of all three types of dietary sugars namely total, free and added sugars. The study team used sugar definitions as adopted by Health Canada, where total sugars includes free and naturally occurring sugars and free sugars includes added sugars and naturally occurring sugars in fruit juices where added sugars include all sugars added to foods during processing or preparation [18].

Data extraction for sugar intakes followed a standard operating procedure to maintain consistency [16]. The dietary data were checked by two data analysts to ensure data quality. Total, free and added sugar intakes were expressed as kcal of sugar per 1000 kcal/day [3]. The study calculations have been described elsewhere [16]. In brief, for this study, total sugar intakes were directly extracted from the ASA24 (Canadian version) nutrient results. Added sugar intakes were manually calculated from ASA24 results using 37 United States Department of Agriculture’s food patterns groups as listed within the Food Patterns Equivalent Database [19]. Added sugars from ASA24 results were converted from teaspoon equivalents to grams and then to kcal, whereas free sugar intakes (including added sugars and 100% fruit juice) were calculated based on whether a listed item contained 100% fruit juice [20].

#### 2.2.3. Predictor Variables (Sociodemographic Characteristics)

Sociodemographic data for the study were gathered from the baseline survey completed by parent 1, who was the first parent to enroll in the study (91% mothers in the overall sample). Sociodemographic characteristics included: child age, child sex, child ethnicity, parent number of years living in Canada, annual household income, parent education and parent marital status. Child age was captured in years and coded as a continuous variable. Child sex was coded as “male” or “female”. Child ethnicity was collected using the question, “How would you describe (child’s name) ethnicity/race?” Due to sparseness in the data across all ethnicities except for white participants, child ethnicity was coded “white” or “other”. Parent number of years in Canada was captured with the question asked as, “How long have you lived in Canada?” For this category, there were 7 response options ranging from “Less than 1 year” to “Greater than 20 years” and this characteristic was coded as a continuous variable by calculating the midpoint of each category. Annual household income was asked as the following question: “What is the total annual income of your household before taxes?” There were 12 response options ranging from <$10,000 to >$150,000 (CAD). The mid-point for each quantitative category was calculated to create a continuous variable for household income. Parent education was asked as, “What is the highest grade or degree you completed in school?” There were 10 response options ranging from “8th grade” or “less to postgraduate training or degree”, which were subsequently collapsed into 3 categories of High school, College or University degree and Postgraduate degree. Parent marital status was assessed using the item, “What best describes your current marital status?” Response options included married, not married but living with a partner, single (never married), divorced, separated, widowed. These categories were subsequently collapsed to “married” (2-parent home- married or common-law) or “not married” (includes single and never married; not married but living with a partner; divorced; separated and widowed).

#### 2.2.4. Data Analysis

Descriptive statistics were used to describe child and parent characteristics. Linear regression models were fitted using generalized estimating equations (GEE) to estimate associations between sociodemographic characteristics including child age, child sex, child ethnicity, household income, parent years in Canada, parent education, parent marital status, and total, free and added sugar. The variable, parent education was coded into 3 categories: high school, university or college and postgraduate degree. A nested modelling approach, performed in R-Studio, was used to globally assess the association between parent education with total, free and added sugar. The GEE approach was utilized to account for siblings in the data [21]. For this study’s data analysis, R [22] was used within RStudio 2021.09.0 Build 351 Version 3.

## 3. Results

### 3.1. Sugar Intake in Preschool-Aged Children and Sociodemographic Characteristics

Table 1 provides information on baseline total, free and added sugar intake in grams/day (mean ± SD) for preschool-aged children, as well as child and parent sociodemographic characteristics. The analytic sample included 267 children from 210 families (Female: 138; Male: 129). The average age of the children was 3.5 ± 1.2 years. There were 205 children (77%) that identified as white and 55 children that identified as other ethnicities. The majority, 177 children, had a parent who was married, and 33 children had a parent who was not married. Household income included 110 families (52%) reporting incomes over $100,000 (CAD).

### 3.2. Associations between Sociodemographic Characteristics and Sugar Intake of Children

A higher total sugar intake was seen in children who were white compared to children of other ethnicities ($\hat{\beta }$ = 31.0; *p* = 0.01; 95% CI = 7.2 to 54.7; Table 2). There was a positive association between child age and free sugar ($\hat{\beta }$ = 8.6; *p* = 0.01; 95%CI = 2.4 to 14.7; Table 3) and added sugar $\hat{\beta }$ = 6.5; *p* = 0.03; 95% CI = 0.8 to 12.2; Table 4) intakes in the study sample. This meant that with every 1-year increase in child age there was an estimated 8.6 kcal increase in free sugar and an estimated 6.5 kcal increase in added sugar intake per 1000 kcal/day. Annual household income was inversely associated with free sugar intake ($\hat{\beta }$ = −2.4; *p* = 0.02; 95% CI = −4.5 to −0.4). No other statistically significant associations were observed. For parent education, there was no association with total sugar (df = 2; *p*-value = 0.77; χ^2^ = 0.53); free sugar (df = 2; *p*-value = 0.05; χ^2^ = 5.9) and added sugar (df = 2; *p*-value = 0.12; χ^2^ = 4.3) in the overall model.

## 4. Discussion

To date, limited research has examined how sociodemographic characteristics may influence young children’s sugar intake. This study found that child age, ethnicity, and annual household income are associated with sugar intake in preschool-aged children. These results advance our fundamental understanding of sociodemographic characteristics that may influence intake of sugar in young children.

Study results suggest that as children age, they eat larger amounts of free and added sugar. These findings are echoed by studies examining overall diet quality and age in children. Furthermore, a study investigating added sugar intake in preschoolers between the ages of 2 to 5 years in North Carolina, USA, found that older children consumed more added sugar than younger children, as older children likely have greater daily caloric intakes [11]. Similarly, across major cohort and longitudinal studies from US, UK, Netherlands and Australia, a recent scoping review found that child age was negatively associated with diet quality [6]. Taken together, these results suggest that interventions in early life should be targeted to parents for children as young as infants, as evidence suggests that intake of added sugars advances with age.

Differences in sugar intake by ethnicity is inconsistent in the research literature. Child ethnicity was found to be a determinant of sugar intakes in the present study and children that identified as white had higher total sugar intake than other ethnic groups. In contrast, a US study of 3-year-old children (*n* = 898) showed that black and Hispanic children consumed greater median intakes of sugar-sweetened beverages (SSB) when compared to white children (*p* < 0.001) [23]. However, the study by Kranz and Siega-Riz, showed that children (*n* = 5652) of Hispanic descent consumed less added sugar than children of other non-Hispanic black or white ethnicities [11]. These differences may be due to the varied racial-ethnic make-up of these American samples compared to our sample from Southwestern Ontario, Canada, which was primarily white, with few children identifying as black or Hispanic. Nevertheless, differences in sugar intake by ethnicity among more diverse racial/ethnic Canadian populations warrants further investigation given its potential impact.

Annual household income was inversely associated with free sugar intake, which is consistent with multiple studies that have highlighted diet-related disparities in relation to income [2,13]. A Canadian study of children aged 4 to 5 years (*n* = 1760), found that children living in low-income neighborhoods had higher intakes of soft drinks and fruit juices (χ^2^ = 14.14, *p* < 0.01) when compared to those living in high-income neighborhoods [8]. Similarly, a Portuguese study investigating multiple SES characteristics found that annual household income was inversely related to children’s intake of SSB and sweets [24]. It is speculated that households with higher income may have access to healthy foods such as fruits and vegetables and that these parents also play an more active role in limiting processed sugary items for their children [7].

The current study found that parent education was not associated with sugar intake in children. However, a Swedish study investigating the dietary pattern of infants at 1 year of age (*n* = 16,070) showed that parent education was inversely associated with the frequency of intake of sweets and pastries [2]. Furthermore, a longitudinal study from the UK found that from 4 to 7 years (*n* = 9550), consumption of a dietary pattern of highly processed foods such as sweets and ice cream was associated with decreasing levels of maternal education [9]. This difference is likely seen in the current study as there was a large proportion (>90%) of parents reporting the attainment of university or postgraduate education. Additional research among more sociodemographic diverse samples in Canada are needed to further elucidate how children’s sugar intake may differ by parental educational level.

## 5. Study Limitations

The present study uniquely contributes to the limited research examining sociodemographic associations with dietary intake in the preschool-age population in the Canadian context, nevertheless, there are some limitations to consider. There could be risk of selection bias during the recruitment of the study sample as families who were motivated to participate in a health-based study could have chosen. There are also many other sociodemographic variables that were not considered in the current study, such as home ownership and number of family members, which could be examined in future studies. Food intake for the study was captured using one 24hour recall, and thus may not reflect usual intake. Parent-reported diet data may be subject to social desirability bias, which could have led to parents underreporting children’s sugar intake. This would have biased results towards the null. Ethnic diversity was limited as 77% of the children were white and over half of the children came from households with incomes greater than $100,000 (CAD). Thus, the study results may not be generalizable to diverse and lower socioeconomic populations and are specific to the families living in the Canadian context.

## 6. Conclusions

This study has found that sugar intake was associated with sociodemographic characteristics of children. Child age was positively associated with free and added sugar intakes and annual household income was inversely associated with free sugar intake. Our data, when considered together with other research exploring these associations, suggest that multiple sociodemographic characteristics play a role in shaping dietary intake of sugar. Thus, investigation of sociodemographic characteristics in larger and more sociodemographic diverse cohorts is warranted. This information can help inform health promotion initiatives when working with varied populations and early intervention programs, through education and awareness among parents to reduce sugar intake in young children.

## Figures and Tables

**Table 1 children-10-01459-t001:** Characteristics of parents and children participating in the Guelph Family Health Study at baseline. Children participants (*n* = 267) and families (*n* = 210).

Type of Sugar *	Daily Intake (g, Mean ± SD) *
Total Sugar	84.1 ± 33.9
Free Sugar	37.4± 26.8
Added Sugar	33.4 ± 23.5
**Characteristic**	**Number (%)**
**Child**	
Male	129 (48.3)
Female	138 (51.7)
*Age, years, mean ± standard deviation (SD)*	3.5 ± 1.2 years
*Ethnicity*	
White	205 (76.8)
Other **	55 (20.6)
**Parent**	
*Parent living for >20 years in Canada*	
Yes	184 (88.0)
No	25 (12.0)
*Household income (CAD)*	
<$60,000	33 (15.7)
$60,000 to $99,999	56 (26.7)
$100,000 to $149,000	63 (30)
≥$150,000	47 (22.4)
*Parent education*	
High school graduate	13 (6.2)
University and College graduate	94 (44.8)
Postgraduate training	103 (49)
*Parent marital status*	
Married	177 (84.3)
Not Married	33 (15.7)

* Average intakes in grams of total, free and added sugar in preschool-aged children. ** Other ethnicities included West Asian (Arab, Iranian, Afghan); Latin American; Chinese; South Asian (East Indian, Pakistani, Sri Lanken); Korean or Japanese; Southeast Asian (Vietnamese, Cambodian, Filipino, Malaysian, Laotian) and Aboriginal/First Nations peoples.

**Table 2 children-10-01459-t002:** Association between child age, child sex, child ethnicity, parent years living in Canada, household income, parent marital status with total sugar intake in preschool-aged children.

	$\hat{\beta }$	SE	*p*-Value	95% CI Low	95% CI High
Child age *	−3.2	3.9	0.4	−11	4.5
Child sex	9.8	9.7	0.3	−9.2	28.8
Child ethnicity **	31	12.1	0.01	7.2	54.7
Parent years living in Canada *	1.9	1.0	0.05	−0.04	3.9
Annual household income *	−0.5	1.2	0.6	−2.8	1.7
Parent marital status **	−15.1	15.4	0.3	−45.3	15

* Child age, Parent years living in Canada and Annual household income were coded as continuous variables. ** Child ethnicity was coded as white or other; Parent marital status was coded as married or other.

**Table 3 children-10-01459-t003:** Association between child age, child sex, child ethnicity, parent years living in Canada, household income, parent marital status with free sugar intake in preschool-aged children.

	$\hat{\beta }$	SE	*p*-Value	95% CI Low	95% CI High
Child age *	8.6	3.1	0.01	2.4	14.7
Child sex	12	9.2	0.2	−6.0	29.9
Child ethnicity **	13	11.3	0.3	−9.2	35.1
Parent years living in Canada *	1.8	1.0	0.05	−0.04	3.7
Annual household income *	−2.4	1.05	0.02	−4.5	−0.4
Parent marital status **	21.8	15.8	0.2	−9.1	52.7

* Child age, parent years living in Canada and annual household income were coded as a continuous variable. ** Child ethnicity was coded as white or other; parent marital status was coded as married or other.

**Table 4 children-10-01459-t004:** Association between child age, child sex, child ethnicity, parent years living in Canada, household income, parent marital status with added sugar intake in preschool-aged children.

	$\hat{\beta }$	SE	*p*-Value	95% CI Low	95% CI High
Child age *	6.5	2.9	0.03	0.8	12.2
Child sex	11.3	7.7	0.1	−3.7	26.4
Child ethnicity **	12.9	9.9	0.2	−6.5	32.3
Parent years living in Canada *	1.69	0.9	0.06	−0.06	3.4
Annual household income *	−1.5	0.9	0.08	−3.2	0.2
Parent marital status **	7.0	11.7	0.6	−16.1	30

* Child age, parent years living in Canada and annual household income were coded as continuous variables. ** Child ethnicity was coded as white or other; parent marital status was coded as married or other.

## Data Availability

The GFHS welcomes external collaborators. Interested investigators can contact GFHS investigators to explore this option, which preserves participant confidentiality and meets the requirements of our Research Ethics Board, to protect human subjects. Due to Research Ethics Board restrictions, we do not make participant data publicly available.

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
