# Peer review of "Sugar Intake among Preschool-Aged Children in the Guelph Family Health Study: Associations with Sociodemographic Characteristics"

_children, 2023, doi:10.3390/children10091459_

Round 1

Reviewer 1 Report

The aim of this study was to explore the links between sociodemographic factors of both children and parents and their consumption of total, free, and added sugar among preschool-aged children.

The topic of interest, sugar consumption in children, is highly significant due to its association with excessive weight gain and dental cavities. Despite efforts to reduce sugar intake, it remains uncontrolled.

A cross-sectional design was employed using the database of the Guelph Family Health Study (GFHS), an ongoing intervention focused on preventing obesity in families with children aged 1.5 to 5 years. Families were recruited through social media posts, posters, and in-person methods.

One consideration to address is the potential impact of families participating in the project's sugar consumption advertisement. Could it be that the families participating have a particular interest in neglecting their children's health, thus potentially leading to an underestimation of sugar consumption in the study?

Furthermore, since the study involved 267 children from 210 families, families with multiple children may over-represent certain characteristics in the sample. Would it be more appropriate to have only one child per family to address this issue?

Additionally, the use of a 24-hour survey may limit the collection of information, as sugar consumption patterns may vary on holidays. It would be beneficial to know whether the questionnaire was collected on weekends, weekdays, or holidays.

How was the required sample size determined for this study? It would be helpful to understand the methodology used for calculating it.

In Table 1 of the results, it would be valuable to indicate whether there are significant differences between the various options presented, such as sugar type, child's gender, ethnicity, and so on.

In Tables 2, 3, and onwards, the wide fluctuation in confidence intervals suggests that the sample size may not be adequate. This is evident from the presence of highly unstable results.

Regarding the discussion, it is important to note that the findings of this study reflect the characteristics of the specific location in which it was conducted. Therefore, the results have local implications that should be considered when interpreting them. However, they may provide insights for future studies in different populations, as they identify potential associations with certain family characteristics. Nonetheless, it is crucial to evaluate these findings within the limitations of the study.

Author Response

Editor’s comments:

1) Please consider the following points in your revisions: adding full experimental details, presenting completely all the results, and describing a comprehensive background to the research in the introduction section.

Response: Thank you for your comments. We have edited and provided additional details to the Introduction and Study Design, Setting, Participants and Enrollment sections of the manuscript.

Revised text: Lines 71 to 127 have been edited as found below.

“Introduction

Overconsumption of sugar intake in children can diminish diet quality, contribute to high energy intake, and lead to excess weight gain and dental caries [1, 2]. Research studies have indicated that preschool-aged children exceed World Health Organization (WHO) guidelines for free sugar intake [1] and consume free and added sugars from a variety of sources [3, 4]. Excess sugar intake can contribute to an increased risk of chronic conditions such as type 2 diabetes and metabolic syndrome later in life [5].

It is widely recognized that early life intervention strategies are critical for establishing healthy eating habits that can continue throughout life. Investigating the various environmental factors that can contribute to the establishment of dietary patterns is a complex process. Young children’s dietary intake is influenced by numerous factors at the level of the child, parent, household, and place of childcare [5, 6]. In high income countries, sociodemographic characteristics especially of the parents significantly impact the diet quality of children [7]. Research has found that energy intake is higher among children from families with low maternal education, parental unemployment and greater number of siblings [8]. Furthermore, individuals of lower socioeconomic status (SES) may have increased chronic disease burden later in life [9]. A scoping review on socioeconomic determinants of food intake patterns in young children (2 to 5 y) highlighted that the current knowledge base in this area for young children is sparse [7]. Emerging evidence highlights that diet-related disparities attributed to sociodemographic characteristics may appear in children as early as 4 y of age [10]. Investigating sociodemographic characteristics of both children and parents in association with intake of sugar in young children may help inform early life diet interventions, nutrition messaging to parents and the development of nutrition policy focused on dietary sugars [6, 11, 12].

There is limited research on how dietary intake of sugar among young children may differ by sociodemographic characteristics. Income and parental education are major indicators driving high sugar intake in many families that have been extensively studied [13]. However, there is heterogeneity in the results and a wide range of other sociodemographic characteristics that have been shown to influence sugar intake including child ethnicity, child immigration status, child age and sex. A German study found that the intake of sweets (including gummies, chocolate and cookies) was higher among preschool-aged children (ages 3 to 6 y) whose families had immigrated to Germany and was inversely associated with parent education level [14]. Similarly, a study from United Kingdom found an association between toddlers (3 y) consuming sweets, biscuits and chocolate with low maternal education levels and higher financial difficulty [15]. Another study conducted in Canada in children aged 4 and 5 years (n=2114), observed that child sex and age were important considerations when investigating sociodemographic differences in sugar intake in young children [8]. Specifically, boys from low SES consumed higher servings of regular soft drinks when compared to girls; also, children aged 5 years were more likely to consume regular soft drinks when compared with children aged 4 years [8]. Considered together these studies present a multifaceted view of numerous sociodemographic factors at both the parental and child level that shape sugar intake in young children.

Most studies have focused on sugar-sweetened beverage (SSB) consumption in relation to sociodemographic [7, 8]. Given the wide range of sources of sugar in children’s diets beyond just SSBs, exploring overall sugar intake, including free and added sugars, may provide a more complete picture of associations between sociodemographic characteristics and children’s sugar intake.  Moreover, there is high variability in the sociodemographic characteristics examined to-date, where some studies have focused solely on parent-level determinants whereas others have explored  child-level determinants [7]. Thus, it is crucial to examine multiple sociodemographic characteristics, including both child and parent-level characteristics, to understand their impact on dietary patterns, specifically sugar intake in young children. These results may help to identify which families to target for interventions designed to reduce children’s sugar intake. Therefore, this study investigated the associations between child and parent sociodemographic characteristics and intakes of total, free and added sugar among preschool-aged and young children”.

Response: Under 2.1 Study Design, Setting, Participants and Enrollment in lines 133-134, the following has been added to the GFHS study description, “This study began in 2017 and is ongoing and includes 246 families at baseline [17]” and in lines 139 to 140 the following is the revised text “The families were given grocery gift cards to recognize their time for participating in the study’.

2) Second, please reduce your repeated rate according to the iThenticate report.  I enclose the iThenticate report in the attachment.

Response: Thank you. We have reviewed the report and noted that overlap pertained to a previously presented scientific abstract of the same results.

Revised text: The abstract has been edited completely from lines 28 to 49.

“Abstract: Background: It is crucial to develop strategies targeted to promote healthy eating patterns in vulnerable populations, especially young children from diverse sociodemographic groups. Thus, this study aimed to investigate the associations between child age, child sex, child ethnicity, parent number of years living in Canada, annual household income, parent education and parent marital status with total, free and added sugar intakes in young children. Methods: This cross-sectional study was a secondary analysis of data gathered in the Guelph Family Health Study. The study included 267 children (129M; 138F) aged 1.5 to 5 years from 210 families. Parents completed questionnaires for children on sociodemographic characteristics and an online 24-hr diet recall. The associations between sociodemographic characteristics and sugar intakes were determined using generalized estimating equations applied to linear regression models. Results: The mean age of the children was 3.5 ± 1.2 years (SD). As children’s age increased, there were higher intake of free and added sugars ( =8.6, P=0.01, 95% CI=2.4 to 14.7 and =6.5, P=0.03, 95% CI = 0.8 to 12.2, respectively). Children who identified as white had a higher total sugar intake than children of other ethnicities ( =31.0, P=0.01, 95% CI=7.2 to 54.7). Additionally, higher annual household income was associated with lower was free sugar intake in children ( = -2.4, P=0.02, 95% CI= -4.5 to -0.4). Conclusions: This study underscores the significant influence of multiple sociodemographic characteristics on sugar intake in young children, providing valuable insights for public health policy and nutrition interventions.  Moreover, this study highlights the need for early behaviour interventions focusing on reducing sugar intake in young children while considering sociodemographic factors.”

3) Third, we found the ethical committee name and code in your article, but we can't find the ethical date. Could you provide us the date as soon as possible?

Response: The approval date, August 15, 2017, has been included in the revised text found on Line 348 under Institutional Review Board Statement.

Reviewer 1:

Comments and Suggestions for Authors

The aim of this study was to explore the links between sociodemographic factors of both children and parents and their consumption of total, free, and added sugar among preschool-aged children.

The topic of interest, sugar consumption in children, is highly significant due to its association with excessive weight gain and dental cavities. Despite efforts to reduce sugar intake, it remains uncontrolled.

A cross-sectional design was employed using the database of the Guelph Family Health Study (GFHS), an ongoing intervention focused on preventing obesity in families with children aged 1.5 to 5 years. Families were recruited through social media posts, posters, and in-person methods.

1. Reviewer#1: One consideration to address is the potential impact of families participating in the project's sugar consumption advertisement. Could it be that the families participating have a particular interest in neglecting their children's health, thus potentially leading to an underestimation of sugar consumption in the study?

Response: Thank you for your comment.   Participants were not recruited specifically for a study on sugar, but we acknowledge the potential for selection bias. This is a secondary analysis of a long-term cohort family study to identify determinants of health. Recruitment was untargeted (i.e., there were no health or diet-based inclusion criteria) and open to families that were contacted through several agencies including early years centres, YMCA, libraries, shopping centres and physician offices. By using a broad recruitment approach and working to engage families from across the socio-economic spectrum and with a range of BMI from healthy to overweight participants, we aimed to reduce our risk of selection bias.  Despite these efforts, we acknowledge some risk of selection bias is possible and added this point in the study limitations.

Revised text: In lines 312 to 314, the following text has been added: “There could be risk of selection bias during the recruitment of the study sample as families who were motivated to participate in a health-based study could have chosen”.

2. Reviewer#1: Furthermore, since the study involved 267 children from 210 families, families with multiple children may over-represent certain characteristics in the sample. Would it be more appropriate to have only one child per family to address this issue?

Response: Our study analysis used generalized estimating equations (GEE) to account for any dependence between sibling participants. Thus, through these statistical analyses, we were able to include all available data on the participants in the study analysis and therefore, we have not made any changes to this into our manuscript.

Reference: Liang KY, Zeger SL. Longitudinal data analysis using generalized linear models. Biometrika 1986;73:13-22.

3. Reviewer#1: Additionally, the use of a 24-hour survey may limit the collection of information, as sugar consumption patterns may vary on holidays. It would be beneficial to know whether the questionnaire was collected on weekends, weekdays, or holidays.

Response: We recognize that we have included only one 24-hour diet recall. The ASA24 is a valid and useful instrument to study associations. The 24-hour recall within ASA24 was completed online for different days by different participants.  When the survey links were sent out to the participants, they were asked to complete the 24-hr recall on a day that represented their typical eating routine. Please see revised text below.

Revised text: Lines 145 to 148 have been edited in the manuscript. “Dietary data were obtained through one 24-hr recall using the Automated Self-Administered 24-hr Dietary Assessment Tool (ASA24)- Canadian version completed by one parent [18]. Parents of the children completed the 24-hour diet recall online and were instructed to complete the recall for a typical day for their child(ren)”.

4. Reviewer#1: How was the required sample size determined for this study? It would be helpful to understand the methodology used for calculating it.

Response: A sample size calculation was not performed as this study was secondary analysis of data collected from the ongoing GFHS.

5. Reviewer#1: In Table 1 of the results, it would be valuable to indicate whether there are significant differences between the various options presented, such as sugar type, child's gender, ethnicity, and so on.

Response: We interpret this comment as a suggestion to stratifying the study data results based on sugar type, gender and ethnicity. This is an excellent suggestion but as a reminder, these variables were treated as covariates to account for any contribution in our statistical analyses.

6. Reviewer#1: In Tables 2, 3, and onwards, the wide fluctuation in confidence intervals suggests that the sample size may not be adequate. This is evident from the presence of highly unstable results.

Response: Thank you for your comment. Our study team recognizes that a wide confidence interval (CI) indicates uncertainty, and that this could cause a lower precision in estimating the true values of the associations between sociodemographic characteristics and sugar intake in young children. We agree that wide CI could be due to a small sample size, and we have addressed this concern in our Conclusions section where we indicate that further research is warranted to replicate these findings with larger cohorts. Despite the limitations of a small sample size and wide CI, our study is elucidating the important role that sociodemographic characteristics of parents and children play in patterns of sugar consumption of young children and filling a significant knowledge gap.

7. Reviewer#1: Regarding the discussion, it is important to note that the findings of this study reflect the characteristics of the specific location in which it was conducted. Therefore, the results have local implications that should be considered when interpreting them. However, they may provide insights for future studies in different populations, as they identify potential associations with certain family characteristics. Nonetheless, it is crucial to evaluate these findings within the limitations of the study.

Response: Thank you for your suggestion. In our Study Limitations section, we have indicated that this study has been conducted in the Canadian context due to limited research on this topic. We have included this further into the revised text below.

Revised text: As per the reviewer’s suggestion, in lines 321 to 322 we have added the following: “Thus, the study results are specific to the Canadian context and may not be generalizable to diverse and lower socioeconomic populations and are specific to the families living in the Canadian context”.

Reviewer 2 Report

Manuscript ID: children-2501425.

This study by Mahajan and colleagues describes sociodemographics characteristics associated with sugar intake among children.

The strenght of the study is the methodology and the diferentation between differents types of sugar. Nevertheless, there are some considerations the authors need to review.

MAJOR COMMENTS:

- Also parent education was not associated with sugar intake in general, it should be take into account that parent post graduate degree have some relation with free sugar intake. Consider explain this data.

- Other limitations of the study are:

- There are other sociodemographics characteristics that are not included: number of members of family, type of house, chronic diseases among parents, smoking, physical activity of parents.

- Please, consider to add information in results regarding total caloric intake and % of sugar. Also, it would be interesting to know main groups of foods that have total sugar intake (sweets, beverage, etc)..

MINOR COMMENTS:

- Line 237: there is a double space between ¨that ¨   and ¨from¨…

- Line 239: check if there is a double space between education and (9).

Author Response

Reviewer 2:

Comments and Suggestions for Authors

This study by Mahajan and colleagues describes sociodemographic characteristics associated with sugar intake among children.

The strength of the study is the methodology and the differentiation between different types of sugar. Nevertheless, there are some considerations the authors need to review.

MAJOR COMMENTS:

1. Reviewer#2: Also parent education was not associated with sugar intake in general, it should be take into account that parent post graduate degree have some relation with free sugar intake. Consider explain this data.

Response: Thank you for your comment. We have added some revised text to highlight these results in the Discussion section.

Revised text: The revised text is shown in lines 302 to 306. “This difference is likely seen in the current study as there was a high proportion (>90%) of parents reporting the attainment of university or postgraduate education. Additional research among more sociodemographic diverse samples in Canada are needed to further elucidate how children’s sugar intake may differ by parental educational level”.

2. Reviewer#2: Other limitations of the study are: There are other sociodemographic characteristics that are not included: number of members of family, type of house, chronic diseases among parents, smoking, physical activity of parents.

Response: We agree that there are other sociodemographic variables. These have been pointed out as additional limitations in the Discussion section.

Revised text: The following text appears in the Study Limitations section on Lines 313 to 315 as follows, “There are also many other sociodemographic variables that were not considered in the current study, such as home ownership and number of family members, which could be examined in future studies.”

3. Reviewer#2: Please, consider to add information in results regarding total caloric intake and % of sugar. Also, it would be interesting to know main groups of foods that have total sugar intake (sweets, beverage, etc).

Response: Thank you for your suggestion. This query has been addressed in separate research studies from the GFHS that previously investigated free sugar from snacks and beverages (Yu, Mahajan et al., 2023) and sugar consumption patterns in young children along with sources of sugar (Mahajan, Yu et al., 2021). Please see references below.

Yu, J., Mahajan, A., Darlington, G., Buchholz, A. C., Duncan, A. M., Haines, J., Ma, D. W. L., & Guelph Family Health, S. (2023). Free sugar intake from snacks and beverages in Canadian preschool- and toddler-aged children: a cross-sectional study. BMC Nutr, 9(1), 44. https://doi.org/10.1186/s40795-023-00702-3

Mahajan, A., Yu, J., Hogan, J. L., Jewell, K., Carriero, A., Annis, A., Sadowski, A., Darlington, G., Buchholz, A. C., Duncan, A. M., Haines, J., Ma, D. W. L., & Guelph Family Health, S. (2021). Dietary sugar intake among preschool-aged children: a cross-sectional study. CMAJ Open, 9(3), E855-E863. https://doi.org/10.9778/cmajo.20200178

MINOR COMMENTS:

4. Reviewer#2: Line 237: there is a double space between ¨that ¨   and ¨from¨…

Response: Thank you for your comment. This has been addressed in line 301.

5. Reviewer#2: Line 239: check if there is a double space between education and (9).

Response: Thank you for your comment. This has been addressed in line 303.

Reviewer 3 Report

The article is well prepared. It presents interesting results on differences in sugar intake in a group of young children by socio-demographic characteristics. The rationale presented in the Introduction is satisfactory, also the presentation of the results and their discussion.

Detailed comments:

1/ The methodology needs clarification regarding parent 1 - "sociodemographic data for the study were gathered from the baseline survey completed by parent 1, who was the first parent to enroll in the study." It is important to know who parent 1-woman or man-was, and perhaps also what his/her involvement in feeding the family/child was.  This could have been important in reporting food intake.

2/ What were the parameters of the linear regression models beyond those given in the tables, such as the percentage of variance explained, i.e. adjusted R2.

Author Response

Reviewer 3:

Comments and Suggestions for Authors

The article is well prepared. It presents interesting results on differences in sugar intake in a group of young children by socio-demographic characteristics. The rationale presented in the Introduction is satisfactory, also the presentation of the results and their discussion.

Detailed comments:

1. Reviewer#3: The methodology needs clarification regarding parent 1 - "sociodemographic data for the study were gathered from the baseline survey completed by parent 1, who was the first parent to enroll in the study." It is important to know who parent 1-woman or man-was, and perhaps also what his/her involvement in feeding the family/child was. This could have been important in reporting food intake.

Response: For this study n=218 mothers completed the ASA24 questionnaires online versus 24 fathers. We have included this as a percentage in the manuscript text to reflect that these were completed mostly by participants mothers that could also impact food intake.

Revised text: We have added the following in lines 174 “Sociodemographic data for the study were gathered from the baseline survey completed by parent 1, who was the first parent to enroll in the study (91% mothers in the overall sample)”.

2. Reviewer#3: What were the parameters of the linear regression models beyond those given in the tables, such as the percentage of variance explained, i.e., adjusted R2.

Response: Thank you for your suggestion. The goal of our analysis was to investigate associations and not to develop a prediction model, therefore R square values are not relevant. Furthermore, the R squares are exceptionally small (close to 0). Therefore, for these reasons the R square values have not been reported in the manuscript.
